# Mapping Utility Poles in Aerial Orthoimages Using ATSS Deep Learning Method

**DOI:** 10.3390/s20216070

**Published:** 2020-10-26

**Authors:** Matheus Gomes, Jonathan Silva, Diogo Gonçalves, Pedro Zamboni, Jader Perez, Edson Batista, Ana Ramos, Lucas Osco, Edson Matsubara, Jonathan Li, José Marcato Junior, Wesley Gonçalves

**Affiliations:** 1Faculty of Engineering, Architecture and Urbanism and Geography, Federal University of Mato Grosso do Sul, Campo Grande 79070900, Brazil; matheusmbg.eng@gmail.com (M.G.); mail.pedrozamboni@gmail.com (P.Z.); jaderluc@gmail.com (J.P.); edson.ufms@gmail.com (E.B.); wesley.goncalves@ufms.br (W.G.); 2Faculty of Computer Science, Federal University of Mato Grosso do Sul, Campo Grande 79070900, Brazil; jonathan.andrade@ufms.br (J.S.); diogo.goncalves@ufms.br (D.G.); edsontm@facom.ufms.br (E.M.); 3Post-Graduate Program of Environment and Regional Development, University of Western São Paulo, Presidente Prudente 18067175, Brazil; anaramos@unoeste.br (A.R.); lucasosco@unoeste.br (L.O.); 4Department of Geography and Environmental Management and Department of Systems Design Engineering, University of Waterloo (UW), Waterloo, ON N2L3G1, Canada; junli@uwaterloo.ca

**Keywords:** object detection, convolutional neural network, utility pole detection

## Abstract

Mapping utility poles using side-view images acquired with car-mounted cameras is a time-consuming task, mainly in larger areas due to the need for street-by-street surveying. Aerial images cover larger areas and can be feasible alternatives although the detection and mapping of the utility poles in urban environments using top-view images is challenging. Thus, we propose the use of Adaptive Training Sample Selection (ATSS) for detecting utility poles in urban areas since it is a novel method and has not yet investigated in remote sensing applications. Here, we compared ATSS with Faster Region-based Convolutional Neural Networks (Faster R-CNN) and Focal Loss for Dense Object Detection (RetinaNet ), currently used in remote sensing applications, to assess the performance of the proposed methodology. We used 99,473 patches of 256 × 256 pixels with ground sample distance (GSD) of 10 cm. The patches were divided into training, validation and test datasets in approximate proportions of 60%, 20% and 20%, respectively. As the utility pole labels are point coordinates and the object detection methods require a bounding box, we assessed the influence of the bounding box size on the ATSS method by varying the dimensions from 30×30 to 70×70 pixels. For the proposal task, our findings show that ATSS is, on average, 5% more accurate than Faster R-CNN and RetinaNet. For a bounding box size of 40×40, we achieved Average Precision with intersection over union of 50% ($AP_{50}$) of 0.913 for ATSS, 0.875 for Faster R-CNN and 0.874 for RetinaNet. Regarding the influence of the bounding box size on ATSS, our results indicate that the $AP_{50}$ is about 6.5% higher for 60×60 compared to 30×30. For $AP_{75}$, this margin reaches 23.1% in favor of the 60×60 bounding box size. In terms of computational costs, all the methods tested remain at the same level, with an average processing time around of 0.048 s per patch. Our findings show that ATSS outperforms other methodologies and is suitable for developing operation tools that can automatically detect and map utility poles.

## 1. Introduction

The power distribution network is one of the most important assets in our daily life. However, the utility poles are vulnerable to situations that lead to deterioration, e.g., extreme weather events [1], corrosion [2] and car accidents [3]. Fragile utility poles can lead to an interruption in the service, pedestrian injury and vehicle and property damages, generating more expenses for utility companies. In response, utility companies conduct constantly human inspections and maintenance in loco to prevent interruption of the service. For that, these companies maintain georeferenced databases with network assets. However, with the network increase, maintaining these databases becomes a challenging and costly task. Therefore, there is a necessity for methods less costly and that can cover greater areas to mapping these assets, such as the utility poles.

Previous research used images or 3D points clouds captured with devices mounted in terrestrial mobile mapping systems to automatically map the utility poles using machine learning methods. Lehtomäki et al. [4] proposed a clustering-based method to find clusters on points clouds obtained from a laser scanning sensor mounted on top of the vehicle. A cluster is classified as a utility pole when several criteria based on the shape of the object are met. A challenge for this method are oblique poles and tree trunk objects which lead to false positives detection. Sharma et al. [5] also proposed a clustering based method such as Mean Shift algorithm to segment utility pole objects from cameras and smartphones images in rural areas, where there are vehicle access restrictions. Most of the machine learning methods are shape-based methods [4,5,6] to find specifically pole-like shapes. However, due to a variety of utilities poles, distance of objects and occlusions make the parameters setting of these algorithms critical. To cope with these challenging tasks, deep learning methods have been proposed in remote sensing applications.

Deep learning methods have been applied significantly to remote sensing image analysis, mainly using convolutional neural networks (CNNs) architectures [7,8,9,10,11,12,13,14]. The use of object detection methods is emerging in the remote sensing community [15]. Zhang et al. [16] applied an object detection deep learning method named RetinaNet on images from Google Street View to utility poles detection. The authors also used a modified brute-force-based line-of-bearing to estimate the positional coordinates. Tang et al. [17] proposed a semi-supervised method that requires only image-level labels (containing power line, positive image, or not, negative image) to detect and localize distribution lines and utility poles. Two Inception-v3 networks were applied in the images obtained from a car-mounted street-level camera pointed upward view at the daytime sky. Nevertheless, in this side-view street images methods the information of the objects needs to be estimated by considering multiple points of view [16], making this method a time-consuming activity mainly in larger areas due to the need for the street by street surveying.

Remote sensing sensors onboard aerial and orbital platforms appear as an alternative enabling the cover of larger areas. Orbital images have a maximum decimetric ground sample distance (GSD), making the detection of utility poles difficult. On the other hand, aerial sensors provide higher spatial resolution imageries. Indeed, the covered area on aerial images is bigger than street-level and it is also possible to observe the distance between utility poles and the interconnection of its lines power distribution. In addition, they can capture the images on heavy traffic roads and can save manpower and time [1]. However, for this application, a few works have been considering aerial images in utility poles mapping, which are commonly used on several remote sensing applications [18].

Alam et al. [1] proposed a variant of SegNet initialized with the pre-trained VGG16 network to detect utility poles from UAV (unmanned aerial vehicle) imagery. The images were captured in urban scenarios flying near the utility poles (almost similar to vertical images captured from vehicles). The goal is to detect the utility pole with SegNet, estimate the angle of detected poles with Hough Transform and predict the pole resilience with the SVM (Support Vector Machine) algorithm. Liu et al. [19] proposed a method to detect utility poles from UAV images with two Faster-RCNN also initialized with the VGG16 network. First Faster R-CNN detects utility poles and the second one crop on the top half of these detected objects to detect the cap pole missing. Moreover, RetinaNet outperformed other deep learning methods in several remote sensing applications [15,20,21]. Despite these initial efforts, there is a lack of research focusing on utility poles automatic mapping using aerial images, mainly using novel deep learning methods, such as Adaptive Training Sample Selection (ATSS). ATSS [22] is a state-of-art method, and it has not been investigated in remote sensing applications.

ATSS proposes a new strategy for selecting negative and positive samples for training object detection methods. Different from methods that use fixed IoU threshold for selecting positive and negative samples such as Faster-RCNN and RetinaNet, ATSS proposes an algorithm to obtain an individual IoU threshold for each ground-truth bounding box. Thus, this approach could improve the detection of small objects or objects that occupy a small area of the bounding box such as the utility poles. Furthermore, this proposal benefits the detection of objects of the same class on different scales and shapes, differently from most of currently state-of-art methods. According to the results shown by Zhang et al. [22], ATSS outperformed current state-of-art methods on the COCO dataset. Despite these results, ATSS has not been investigated in remote sensing applications.

Developing automatic methods to detect poles is crucial to maintain the sustainability of the growing power distribution network, face the increase of extreme weather events and maintain the quality of service. The objective of this study was to evaluate the performance of ATSS to detect and map utility poles in aerial images. We compared the achieved results with Faster R-CNN [23] and RetinaNet [24], which are common methods applied in remote sensing image analysis. RetinaNet, for example, outperformed other deep learning methods in several remote sensing applications [15,20,21].

Most research using aerial images is focused on utility poles inspection, which requires low flights for monitoring their elements. For mapping purposes higher flights are preferable, making it difficult to identify the poles in the images, as can be verified in Figure 1. This leads to some challenges during the detection, such the presence of shades, poles occluded by other objects, different backgrounds in the images, different types and sizes of poles and the fact that poles represent a small portion of the bounding box.

The rest of this paper is organized as follows. Section 2 demonstrates the method developed here. Section 3 presents the results obtained with our approach while comparing them with other methods. Section 4 discusses the implications of the results of our approach. Section 5 concludes our research.

## 2. Material and Methods

### 2.1. Study Area

We used as a study area the urban area of the city of Campo Grande, in the state of Mato Grosso do Sul, Brazil (Figure 2).

The aerial RGB orthoimages were provided by the city hall of Campo Grande, state of Mato Grosso do Sul, Brazil. The orthoimages have a ground sample distance (GSD) equal to 10 cm. In total, 1057 orthoimages with dimensions 5619 × 5946 pixels were used in the experiments. These images were split in 99,473 patches, where 111,583 utility poles were identified as ground-truth. Details regarding the experimental setup are presented in Section 2.3.

### 2.2. Pole Detection Approach

In general, the labels provided for the poles consist of their coordinates due to georeferenced databases with network assets. The dataset was manually labeled with one point (represented by two values) at the base of each pole. Since object detection methods need a bounding box instead of a point, each ground-truth label (point) was converted to a rectangle (represented by four values). To do that, a rectangular box (well know as bounding box) centered at this point was built by setting its width size $w_{b}$ and height size $h_{b}$. In this way, the midpoint of each bounding box is represented by the point (ground-truth label) to determine bounding box localization and the two parameters $w_{b}$ and $h_{b}$ determine its size. In the experiments, we evaluated the influence of different sizes ($w_{b}$ and $h_{b}$ values) on the results. Figure 3 shows the labels represented by a coordinate and the corresponding bounding boxes.

Given the annotated bounding boxes, the object detection algorithms can be trained to learn these bounding boxes and be able to predict the bounding boxes’ location in an image. In this sense, the state-of-art algorithms such as RetinaNet [24] and Faster R-CNN [23] usually generate a large number of boxes (named as anchor boxes), estimate how good are these predicted anchors with regards to the bounding boxes and split them into positive and negative examples. The quality of anchor boxes is estimated considering the ratio of intersection to the union between anchor box and bounding boxes, and this measure is well known as Intersection over the Union (IoU). Since refining these large number of generated anchors, which have four parameters (spatial coordinates) each, can be costly, recently anchor-free methods have emerged [25]. However, Zhang et al. [22] concluded that the main impact on the method’s performance is not only considering anchor-based and anchor-free representations, but how they select positive and negative samples. The main idea of the ATSS [22] algorithm is to select a small set (top *K*) of good anchors boxes based on object’s statistical features for classification and regression steps.

We compared the ATSS method with RetinaNet and Faster R-CNN methods. The object detection methods used in this study are briefly described as follows. The methods were implemented based on the source codes proposed by Multimedia Laboratory, via the MMDetection project. The open-source project is available at https://github.com/open-mmlab/mmdetection.

Faster R-CNN [23] is a convolutional neural network (CNN) used for detecting objects in two stages. The CNN architecture can be described into three components named as backbone (aimed to extract feature maps from images), neck (refine the feature maps) and head (stands for classification and prediction tasks, for example) [26]. Using features maps generated by a backbone (ResNet50 in this work), the first stage, named Region Proposal Network (RPN) at the neck component, receives the features maps and outputs as result candidates object bounding boxes. The second stage also receives the features maps and aim through a Region of Interest Pooling layer to extract features from each of the candidate bounding boxes proposed by the first stage. This operation is based on max pooling, aiming to obtain a fixed-size feature map that does not depend on the dimensions of the bounding boxes proposed by the RPN. Finally, a softmax layer predicts the location and classes of the proposed regions at the head component.

RetinaNet [24] is a one stage object detector that solves the class imbalance by reducing the loss attributed to well-classified images. The class imbalance occurs when the number of background examples is much larger than the examples of the object of interest. Through a new loss function, training is focused on concrete examples and prevents the high number of background examples from making it difficult for the network to learn the method. The RetinaNet architecture is based on a backbone, a neck and two task-specific subnetworks (at head component). The backbone and neck consist of ResNet-50 (in this work) and a Feature Pyramid Network from [27], respectively, responsible for obtaining a feature map across an entire input image. The first subnet is a small Fully Convolution Network, a simple network with five convolutional layers that are attached to the neck component. This subnet is responsible for predicting, at each spatial position throughout the image, a probability of the object’s presence. The second subnet has an identical structure to the first, but it is responsible for calculating the regression to estimate the bounding box coordinates. Both the first and second subnets are parallel.

ATSS (Adaptive Training Sample Selection) [22] is a method that introduces a new way of defining positive and negative samples during the training phase. Unlike other state-of-the-art methods that have a high number of hyperparameters, ATSS has only one hyperparameter *K*, corresponding to the number of anchors closest to the center of the ground-truth that are selected from each level of the pyramid feature. This hyperparameter is kept fixed at a value of 9. The ATSS also differs from other state-of-the-art methods by using an individual IoU threshold for each ground-truth during the training, obtained by adding the average and standard deviation of the anchor box IoU’s proposals in relation to the ground-truth. Bounding boxes with IoU greater than the threshold calculated for the respective ground-truth are considered positive samples. In our experiments, the ATSS was set with ResNet-50 as the backbone and FPN as the neck.

### 2.3. Experimental Setup

For our experimental setup, we divided the RGB orthoimages into training, validation and test sets. We used 634 (60%), 212 (20%) and 211 (20%) orthoimages from different areas for training, validation and testing, respectively. Each orthoimage has 5619 × 5946 pixels (561.90 m × 594.60 m). Since the object detection methods receive smaller images as input, each orthoimage was split into non-overlapping patches of 256 × 256 pixels (25.60 m × 25.60 m), covering an area of approximately 655.36 m^2^ per patch. The number of orthoimages and image patches are shown in Table 1.

For the training process, we initialized the backbone of all object detection methods with pre-trained weights from ImageNet (http://www.image-net.org/). We applied a Stochastic Gradient Descent optimizer with a momentum equal to 0.9. For this, we used the validation set to adjust the learning rate and the number of epochs to reduce the risk of overfitting. We empirically assessed learning rates (0.0001, 0.001 and 0.01) and found that the convergence of the loss function is better for 0.001 and stabilized over 24 epochs. Figure 4 illustrates the loss curve for all methods. As we can see, the loss function decreases rapidly in the first iterations and stabilizes at the end, indicating that the number of epochs is sufficient and the learning rate is adequate. After performing the adjustments, we set the initial learning rate to 0.001 and the number of epochs to 24 (∼700 k iterations).

The proposed application was developed using MMDetection framework [26] on the Ubuntu 18.04 operating system. Training and testing procedures were conducted in a computer equipped with an Intel${}^{®}$ Xeon(E) E3-1270 @3.80 GHz CPU, 64 GB of RAM Memory along with Titan V graphics card, a GPU produced by NVIDIA containing 5120 CUDA (Compute United Device Architecture) cores and 12 GB of graphics memory.

### 2.4. Method Assessment

The performance of the methods is evaluated using the Average Precision (AP). To obtain the precision and recall metrics, the Intersection Over Union (IoU) was calculated as the overlapping area between the predicted and the ground-truth bounding boxes divided by the area of union between them. In the experiments, we used common IoU values of 0.5 and 0.75. If the prediction obtains IoU greater than the threshold, the prediction is considered as true positive (TP); otherwise, it is a false positive (FP). A false negative (FN) occurs when ground-truth boxes are not detected by any prediction. Using the metrics described above, precision and recall are estimated using Equations (Equation 1) and (Equation 2), respectively. The area under the precision–recall curve represents the average precision (AP).
(1)$$\begin{array}{c} P=\frac{TP}{TP+FP} \end{array}$$
(2)$$\begin{array}{c} R=\frac{TP}{TP+FN} \end{array}$$

## 3. Results

This section describes the results obtained in our experiments. Section 3.1 shows a quantitative analysis of the result, while Section 3.2 discusses the qualitative ones. Lastly, Section 3.3 reports the computational costs of the assessed methods.

### 3.1. Quantitative Analysis

First, we evaluated the object detection methods using AP with two thresholds. In the first test, the average precision was obtained using a IoU threshold of 0.5 ($AP_{50}$). On the second test, we increased the threshold to 0.75, and the average precision was calculated ($AP_{75}$). Table 2 shows the results for each test using a fixed size bounding box of 40×40.

It is observed that ATSS outperformed the other two methods by about 4% for $AP_{50}$ and 7% for $AP_{75}$. The results show the higher accuracy of ATSS compared to other state-of-the-art architectures. It is also noted that, for $AP_{50}$, Faster R-CNN and RetinaNet have similar results. When the threshold is increased to 0.75, Faster R-CNN achieved slightly better performance. Considering the 111,583 utility poles present in our dataset, the higher accuracy in $AP_{50}$ translates into about 4464 poles identified by ATSS that the Faster R-CNN and RetinaNet methods were unable to detect. In $AP_{75}$, this number rises to 7811 utility poles located by ATSS that the two other methods were unable to detect.

To assess the influence of the bounding box size, Table 3 presents the results using five sizes in the construction of the ground truth. The results were obtained using ATSS as an object detector, since it obtained the best results in the previous experiment. The results indicate that increasing the bounding box size from 30×30 to 60×60 improves $AP_{50}$ by 6.5% and $AP_{75}$ by 23.1%. In terms of detected utility poles, ATSS with 60 × 60 bounding boxes located around 7523 for $AP_{50}$ and 25,776 for $AP_{75}$ more utility poles in comparison with 30 × 30 bounding box size.

### 3.2. Qualitative Analysis

The results were also analyzed from a qualitative point of view. The priority was to analyze the assertiveness of the methods for poles in different conditions such as shaded, covered by trees or in different types of terrain. We also assessed the performance of the methods while detecting different types, sizes and shapes of the utility poles presented in the dataset. The results shown in this section were obtained using $AP_{50}$ and 40×40 bounding box size.

Figure 5 shows the detection of poles occluded by the top of trees. It is observed that the most assertive method for this scenario was ATSS. Faster R-CNN and RetinaNet were able to detect the region of the image in which the object was present, but both methods provided multiple bounding boxes for the same object when it is occluded by trees.

Figure 6 shows the poles that are visible but in a shadow region. Note that the characteristic of the predictions is similar to the previous scenario. When the poles are shaded, Faster R-CNN and RetinaNet did not have a good assertiveness. Several examples were verified in the dataset in these lighting situations, and both methods end up generating multiple bounding boxes for the same object, while ATSS again had no problems locating the utility poles.

Regarding the different sizes and types of poles, Figure 7 shows the detections and the corresponding terrestrial image obtained from Google Street View. ATSS obtained a good performance, detecting utility poles of the most varied types and sizes. Faster R-CNN and RetinaNet methods obtained good assertiveness by processing images of smaller scale poles, such as lighting poles or low voltage electric poles, but failed to locate larger poles, such as the models used for high voltage transmission of electrical energy, shown in the last row of Figure 7.

In the dataset, there are examples of utility poles installed on different types of terrain, such as grass, street, sand, etc. Figure 8 illustrates the behavior of the methods under this variation of background. It was observed that changing the type of terrain is not a factor that interferes with the assertiveness of the methods. All methods achieved good performance when processing images of utility poles installed in different types of terrain.

### 3.3. Computational Costs

Since training and validation were performed on the workstation described in Section 2.3, we also used the same workstation to assess the computational costs of the methods. For that, the processing time of all 19,480 patches of the test set was obtained for each method. Considering that each patch covers an area of 655 m^2^, we also established as a metric the area scan speed, that is the amount of area per second that each method processed. With the total processing time, the average processing time for each patch and the area scan speed was estimated, as detailed in Table 4.

As expected, the three methods achieved an almost equal average processing time, with the RetinaNet method being only 1 ms faster compared to the other methods, a minimal difference that can be considered insignificant. This equality in terms of computational costs is justified by the fact that all three tested methods have ResNet50 architecture as a backbone.

The obtained computational costs demonstrate that the workstation efficiently handled the high number of images in the dataset. The processing of the 19,480 test images took about 15 min for each method. The training and validation steps for each method took around 20 h.

The scan speed obtained demonstrates the great advantage of the use of aerial images compared to the street by street mapping, as described by Zhang [16], with our proposed method scanning in just a few seconds the utility poles of neighborhood-sized areas. The average time shows that the methods have the potential to be applied in real-time and may also be embedded in processors with low computational capacity in the future.

## 4. Discussion

As described in the Section 3, it was observed that ATSS was, on average, 5% more accurate than Faster R-CNN and RetinaNet using bounding box with size 40×40 pixels. Regarding computational costs, the three methods remained at the same level.

The size of the bounding box proved to be an influential parameter in the average precision obtained by ATSS. By increasing the size of the bounding box from 30×30 to 60×60, we identified that $AP_{50}$ increases from 87.9% to 94.4% and from 62.3% to 85.3% for $AP_{75}$. As in the dataset the utility poles are seen from different angles, smaller bounding boxes may not completely cover the object, as shown in Figure 9 with 40×40 size bounding boxes. The use of larger bounding boxes not only eliminates the risk of not fully cover the object of interest, but also allows the method to learn elements inherent to the object that can assist in its detection, such as the shadow of the utility pole on the ground. However, very large bounding boxes can include irrelevant information for object detection as observed for 70×70.

When observing the image dataset, it is noted that pole detection in aerial images is a challenging task, since there are different terrains, sizes and types of utility poles and poles covered by other objects such as trees, among other situations. Another problem is related to the size of each object in the ground-truth. The object itself corresponds to a small portion of the bounding box, where everything else corresponds to the background of the image.

Considering these difficulties, the relative superiority of the ATSS in relation to the other two methods can be credited to the proposal to calculate an IoU threshold for each ground-truth during training. As the utility pole itself occupies a small area of the bounding box, when using a fixed value of IoU threshold for all ground-truths, such as > 0.5 or > 0.75, a sample can be considered positive even if the bounding box intersects only the background presented in the ground-truth, exclusively to the fact that the prediction has a larger IoU than a predetermined value. The calculation of the IoU threshold proposed by the ATSS method avoids the occurrence of this scenario, preventing the method from learning features that do not correspond to the object of interest.

The calculation of IoU threshold for each ground-truth proved to be a powerful tool also for datasets in which the objects have different sizes and shapes, a scenario presented in this work. Obtaining an IoU threshold for each ground-truth enabled the method to learn the features of different pole models. The results obtained demonstrate that calculating a particular IoU threshold for each object during the training contributed to the ATSS method learning more general features of the utility poles present in the dataset.

## 5. Conclusions

We proposed and evaluated the performance of the ATSS deep learning method for the detection and location of utility poles in aerial orthoimages. Two state-of-the-art methods widely used in remote sensing applications, Faster R-CNN and RetinaNet, were used for comparison purposes.

The task was performed satisfactorily by all methods; however, the ATSS obtained the best performance among the three experienced, with 91.3% average precision using IoU threshold of 0.5, being on average 4% more accurate than the Faster R-CNN and RetinaNet methods. This result was expected due to the innovations proposed by the ATSS method regarding other state-of-the-art methods. The computational costs of the three methods were very similar, with average processing of 20.83 images per second. The best bounding box size setup found for the ATSS method was with dimensions of 60×60 pixels. This setup achieved an average precision of 94.4% for $AP_{50}$ and 85.4% for $AP_{75}$, an increase of 6.5% for $AP_{50}$ and 23.1% for $AP_{75}$ when compared with 30×30 bounding boxes.

The results indicate that the ATSS method is a strong candidate for the development of an operational tool for detecting and locating utility poles. In our application it obtained better performance in comparison to other methodologies proposed and published in other papers, such as the location of streetlights by street-level images [16].

## Figures and Tables

**Figure 1 sensors-20-06070-f001:**
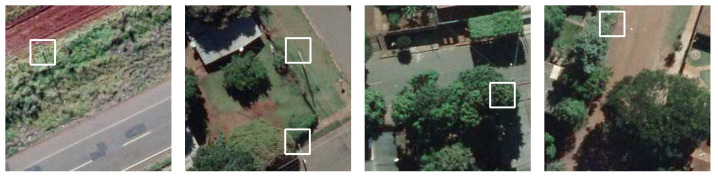
Examples of electric poles present in our dataset.

**Figure 2 sensors-20-06070-f002:**
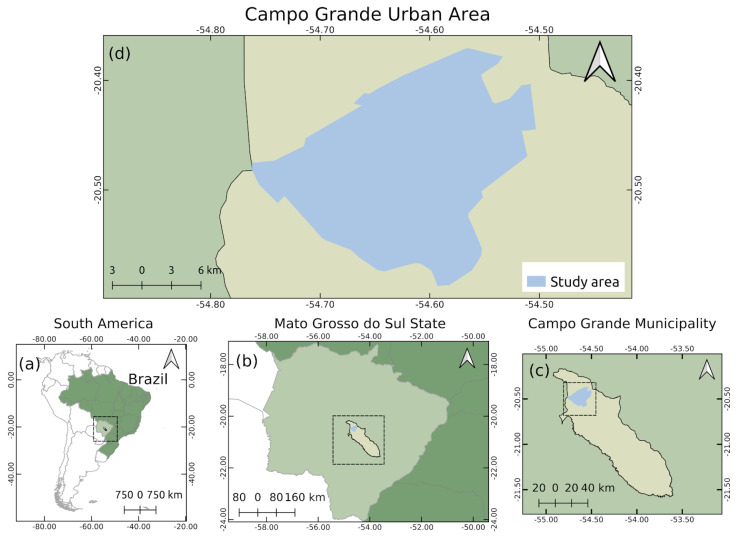
Study area location: (**a**) South America and Brazil; (**b**) Mato Grosso do Sul; (**c**) Campo Grande; and (**d**) Study area.

**Figure 3 sensors-20-06070-f003:**
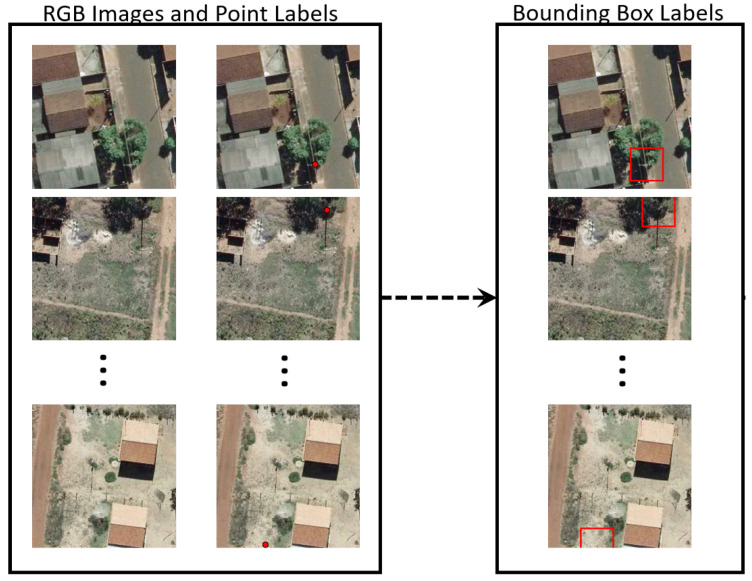
Steps of the pole detection approach.

**Figure 4 sensors-20-06070-f004:**
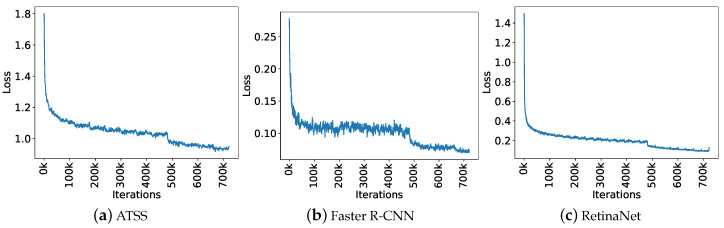
Loss function during training for the three object detection methods.

**Figure 5 sensors-20-06070-f005:**
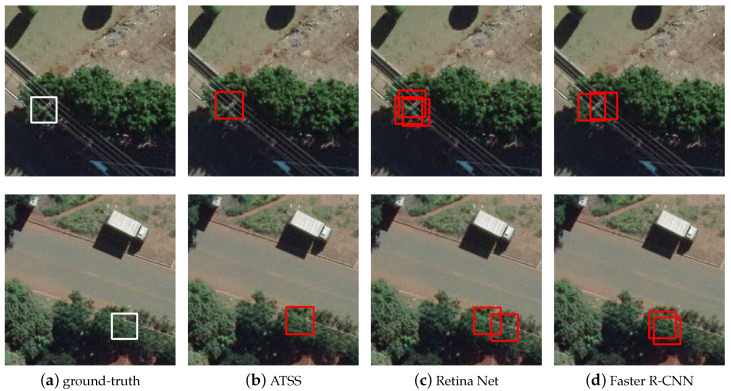
Results obtained by the methods when locating utility poles occluded by trees tops: (**a**) the ground-truth; (**b**) ATSS; (**c**) RetinaNet; and (**d**) Faster R-CNN.

**Figure 6 sensors-20-06070-f006:**
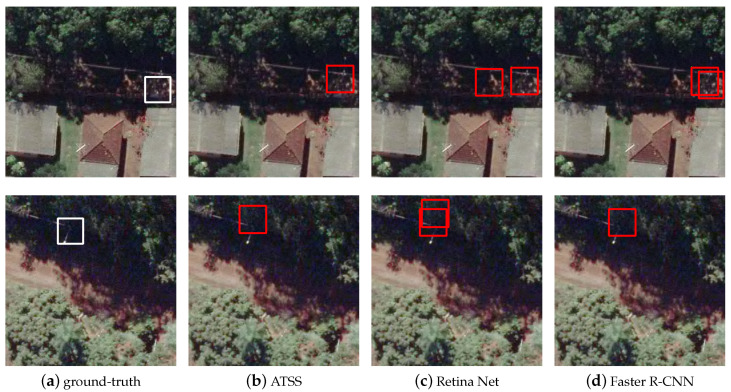
Examples of images with shaded poles: (**a**) ground-truth; (**b**) ATSS; (**c**) RetinaNet; and (**d**) Faster R-CNN.

**Figure 7 sensors-20-06070-f007:**
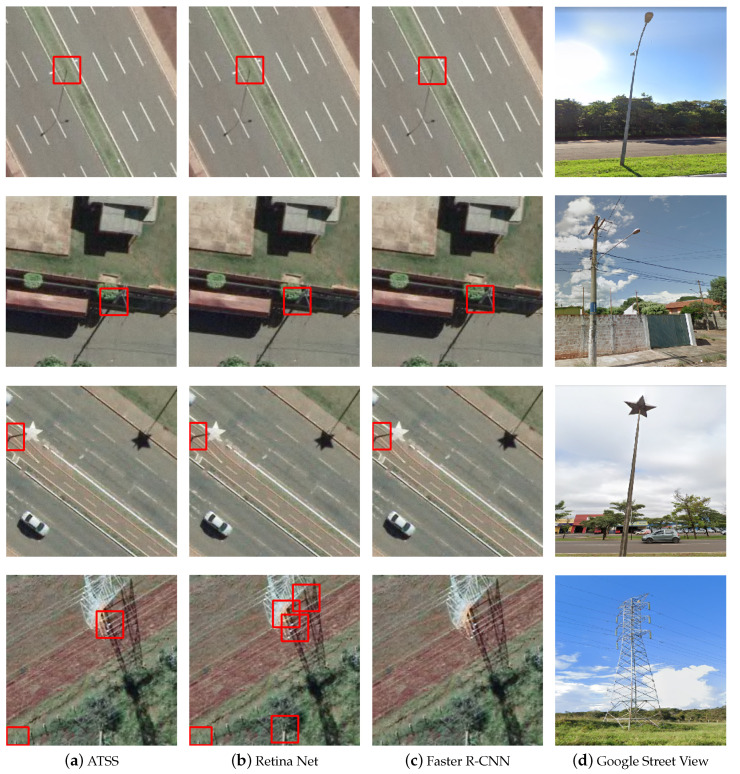
Examples of images with different models and shapes of poles: (**a**) ground-truth; (**b**) ATSS; (**c**) RetinaNet; and (**d**) Faster R-CNN.

**Figure 8 sensors-20-06070-f008:**
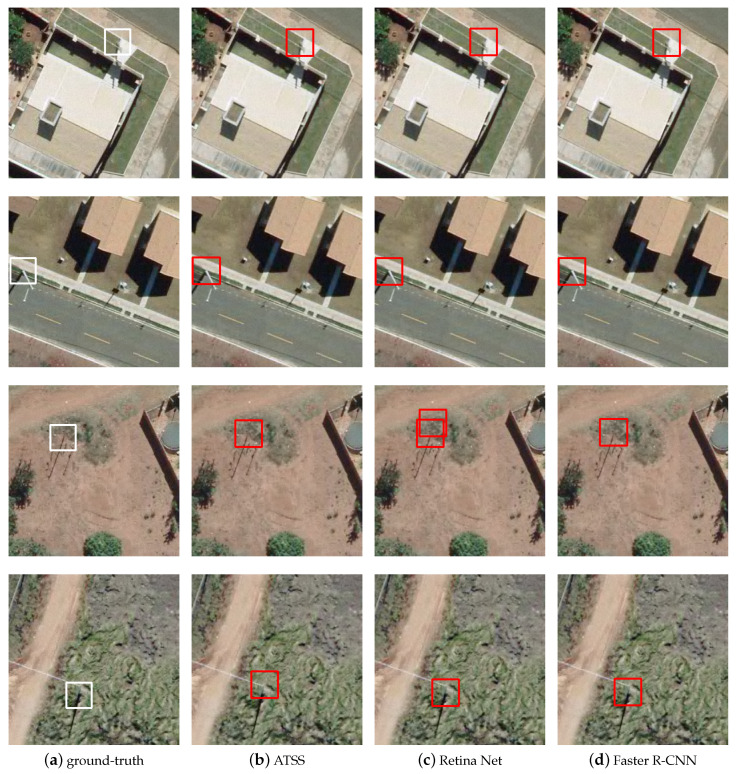
Examples of images with utility poles in different types of terrain: (**a**) ground-truth; (**b**) ATSS; (**c**) RetinaNet; and (**d**) Faster R-CNN.

**Figure 9 sensors-20-06070-f009:**
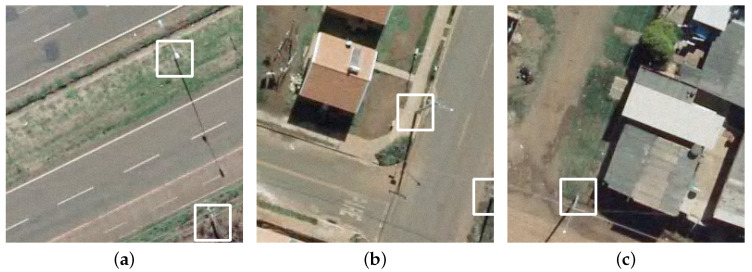
Examples of utility poles that are not fully cover with a 40×40 bounding box size: (**a**) example 1; (**b**) example 2; and (**c**) example 3.

**Table 1 sensors-20-06070-t001:** Description of the training, validation and test sets of the poles dataset.

Dataset	Number of Patches	Number of Orthoimages
Training	60,124	634
Validation	19,869	212
Testing	19,480	211
Total	99,473	1057

**Table 2 sensors-20-06070-t002:** Average precision for the object detection methods. To generate the ground truth, a fixed size bounding box of 40×40 was used.

Method	${AP}_{50}$	${AP}_{75}$
Faster R-CNN	0.875	0.682
RetinaNet	0.874	0.673
ATSS	0.913	0.749

**Table 3 sensors-20-06070-t003:** Average precision for different fixed size bounding boxes using ATSS.

Size	${\mathrm{AP}}_{50}$	${\mathrm{AP}}_{75}$
30×30	0.879	0.623
40×40	0.913	0.749
50×50	0.931	0.815
60×60	0.944	0.854
70×70	0.944	0.853

**Table 4 sensors-20-06070-t004:** Processing time evaluation of the methods.

Method	Average Time (s)	Area Scan Speed (m^2^/s)
Faster R-CNN	0.048	13,651.14
Retina Net	0.047	13,943.82
ATSS	0.048	13,651.14

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
