# Peer review of "Mapping Utility Poles in Aerial Orthoimages Using ATSS Deep Learning Method"

_sensors, 2020, doi:10.3390/s20216070_

Round 1
Reviewer 1 Report
The authors of the article compared the performance of three methodologies of utility poles detection on orthoimages. Advantages of the article: 1) Authors prepared huge data set to compare the methods. 2) Autor selected eligible methods. 3) Authors compared the three methods, i.e. ATSS, R-CNN and RetinaNet. Disadvantages of the article: 1) The compared methods are not novel. 2) Description of the methods is short. 3) Scientific soundness of the article can be easily improved by more citations.
Author Response
The authors of the article compared the performance of three methodologies of utility poles detection on orthoimages. Advantages of the article: 1) Authors prepared huge data set to compare the methods. 2) Autor selected eligible methods. 3) Authors compared the three methods, i.e. ATSS, R-CNN and RetinaNet. Disadvantages of the article: 1) The compared methods are not novel. 2) Description of the methods is short. 3) Scientific soundness of the article can be easily improved by more citations.
Answer: Thank you for your valuable comments. All the modifications in the text are in blue. Regarding the disadvantages:
1. We focused on ATSS, as it is a recent method published in CVPR in 2020. We compared it with Faster RCNN and RetinaNet. We totally agree that Faster RCNN and RetinaNet methods are not novel; however, they are commonly used in remote sensing applications. For example, Li et al. (2020) proposed a benchmark and showed that RetinaNet outperformed several object detection methods. Other recent works also used Faster R-CNN and RetinaNet (Santos et al., 2019; Santos et al., 2020). We modified the abstract and the introduction to make this point more clear, as you suggested.
2. We added more information regarding the methods, as suggested. The following paragraph was added:
Given the annotated bounding boxes, the object detection algorithms can be trained to learn these bounding boxes and be able to predict the bounding boxes’ location in an image. In this sense, the state-of-art algorithms such as RetinaNet[20] and Faster R-CNN[19] usually generate a large number of boxes (named as anchor boxes), estimate how good are these predicted anchors regards to the bounding boxes, and split them into positive and negative examples. The quality of anchor boxes is estimated considering the ratio of intersection to the union between anchor box and bounding boxes,and this measure is well known as Intersection over the Union (IoU). Since refining these large number of generated anchors, which have four parameters (spatial coordinates) each one, can be costly, recently anchor-free based methods have emerged like the ATSS[18]. The main idea of the ATSS algorithm is to select good anchors point (each box is represented by its middle point).
3. We included other references [7-14] in the introduction, as suggested, mainly from 2020. If you have other suggestions, please, inform us with a link. Thank you again.
Reviewer 2 Report
This paper is aimed at proposing and testing a supposedly novel method for the automatic detection of utility poles in urban environments. The following major revisions are mandatory before publication of this study may be envisaged:
1 - The abstract is full of acronyms and not clearly written. What exactly is the problem the study tries to solve? Which type of method exactly is used, and why, by comparison with other approaches? What is the unifying take-home message of this article on the basis of the most striking results obtained by the method proposed?
2 - The first part of the introduction is well written and elaborates on the framework in which this study is couched. However, later on it fails to clearly address the motivation for this study. Simply stating that the objective of this study is to evaluate the method's performance for detecting utility poles in aerial images is not clear enought, i.e. not explicitly justified. Why this particular method, and not any of the others described in the introduction? The referral to bounding boxes with varying fixed sizes and how they are obtained from geolocalization coordinates, or how the RGB images of 256x256 pixels were generated, belongs into the "Materials and Methods" section, NOT in the introduction. The latter needs to state clearly and explicitly why the proposed method can be expected to outperform other ones by comparison, in anticipation of what is then discussed in the Results sections.
3 - In the Materials and Methods section it is stated that the object labels refer to/correspond to coordinates from georeferenced databases accessed (directly) by the network. Then, to identify objects in the procedure as described further on, the labels are converted from coordinates to width x height pixel-dimensional rectangles, for estimating what the authors call "bounding boxes", on which the object detection method is trained.The object detection methods exploited in the study are too briefly described. More detailed information needs to be given to enable readers to understand what was done exactly, and to clarify why in regard to what is elaborated in the introduction.
4 - The authors use a training procedure with pre-trained weights from ImageNet with stochastic gradient descent optimization on a validation set for adjusting learning rates and number of epochs. They then determine a fixed learning rate and number of epochs. The implicit and explicit criteria underlying the choices made here need to be better explained.
5 - The application/method proposed here was developed within the detection framework as stated in the Introduction and in the Material and Methods section using the Linux-equivalent Ubuntu operating system. Training and testing procedures were performed using a single computer workstation. How this station integrates the (heavy) image material for further processing needs to be better explained.
6 - In the Results section, the authors use % performance advantages to assess the superiority of their method in comparison with others. What these percentages mean in terms of statistical effects with probability boundaries is unclear. 4%, 6%, 7% et. all seem quite small, and none of them represent a statistically significant performance advantage. This problem needs to be addressed clearly (i.e.the meaning of the percentages needs to be clarified), and other statistical tests need to be performed to back up the conclusions drawn here.
Author Response
This paper is aimed at proposing and testing a supposedly novel method for the automatic detection of utility poles in urban environments. The following major revisions are mandatory before publication of this study may be envisaged:
Answer: Thank you for your valuable comments.
1 - The abstract is full of acronyms and not clearly written. What exactly is the problem the study tries to solve? Which type of method exactly is used, and why, by comparison with other approaches? What is the unifying take-home message of this article on the basis of the most striking results obtained by the method proposed?
Answer: The abstract was modified following your suggestion. First, we included a description of all the acronyms (ATSS, Faster R-CNN, RetinaNet, and others). We tried to make clear what is our main objective (use of the ATSS method for the detection and mapping of the utility poles in urban environments using top-view images). ATSS is a novel method, published in CVPR 2020, and still not investigated in remote sensing applications. Also, Faster R-CNN and RetinaNet are commonly used methods in remote sensing. We included this information in the abstract. Finally, we added a sentence at the end of the abstract to show the message, like you suggested.
2 - The first part of the introduction is well written and elaborates on the framework in which this study is couched. However, later on it fails to clearly address the motivation for this study. Simply stating that the objective of this study is to evaluate the method's performance for detecting utility poles in aerial images is not clear enought, i.e. not explicitly justified. Why this particular method, and not any of the others described in the introduction? The referral to bounding boxes with varying fixed sizes and how they are obtained from geolocalization coordinates, or how the RGB images of 256x256 pixels were generated, belongs into the "Materials and Methods" section, NOT in the introduction. The latter needs to state clearly and explicitly why the proposed method can be expected to outperform other ones by comparison, in anticipation of what is then discussed in the Results sections.
Answer: Thank you for this observation. First, we removed from the introduction section all the parts related to the methods. Similar to the abstract and with more details, we justified the adoption of the ATSS method and the selection of Faster R-CNN and RetinaNet as comparison methods. In general, this is related to the usage of them in remote sensing applications.
3 - In the Materials and Methods section it is stated that the object labels refer to/correspond to coordinates from georeferenced databases accessed (directly) by the network. Then, to identify objects in the procedure as described further on, the labels are converted from coordinates to width x height pixel-dimensional rectangles, for estimating what the authors call "bounding boxes", on which the object detection method is trained. The object detection methods exploited in the study are too briefly described. More detailed information needs to be given to enable readers to understand what was done exactly, and to clarify why in regard to what is elaborated in the introduction.
Answer: We added more information regarding the methods, as suggested. The following paragraph was added:
Given the annotated bounding boxes, the object detection algorithms can be trained to learn these bounding boxes and be able to predict the bounding boxes’ location in an image. In this sense, the state-of-art algorithms such as RetinaNet[20] and Faster R-CNN[19] usually generate a large number of boxes (named as anchor boxes), estimate how good are these predicted anchors regards to the bounding boxes, and split them into positive and negative examples. The quality of anchor boxes is estimated considering the ratio of intersection to the union between anchor box and bounding boxes,and this measure is well known as Intersection over the Union (IoU). Since refining these large number of generated anchors, which have four parameters (spatial coordinates) each one, can be costly, recently anchor-free based methods have emerged like the ATSS[18]. The main idea of the ATSS algorithm is to select good anchors point (each box is represented by its middle point).
4 - The authors use a training procedure with pre-trained weights from ImageNet with stochastic gradient descent optimization on a validation set for adjusting learning rates and number of epochs. They then determine a fixed learning rate and number of epochs. The implicit and explicit criteria underlying the choices made here need to be better explained.
Answer: To address this issue, we have described the methodology for choosing the learning rate and the number of epochs in the second paragraph of Section 2.3. We have also included the loss function of the three methods (Figure 4) which shows that the training has been stabilized with the chosen learning rate and number of epochs.
5 - The application/method proposed here was developed within the detection framework as stated in the Introduction and in the Material and Methods section using the Linux-equivalent Ubuntu operating system. Training and testing procedures were performed using a single computer workstation. How this station integrates the (heavy) image material for further processing needs to be better explained.
Answer: We added more details in the computational cost section (section 3.3). The area scan speed (m2/s) was included in Table 4. The time to train the models, and also to run the inference in the test set was also included and discussed in the corresponding section. Information regarding the computer workstation was also included.
6 - In the Results section, the authors use % performance advantages to assess the superiority of their method in comparison with others. What these percentages mean in terms of statistical effects with probability boundaries is unclear. 4%, 6%, 7% et. all seem quite small, and none of them represent a statistically significant performance advantage. This problem needs to be addressed clearly (i.e.the meaning of the percentages needs to be clarified), and other statistical tests need to be performed to back up the conclusions drawn here.
Answer: We compared them regarding the number of detected poles to turn the comparison more explicit. We agree that looking only at the percentages it looks quite small, but looking at the number of detected utility poles, that is a different perspective. If we consider 4% or 7% on our dataset, that is a considerable number of utility poles that ATSS located and that RetinaNet and Faster R-CNN did not. The following paragraph was added:
Considering the 111583 utility poles present in our dataset, the higher accuracy in AP50 translates into about 4464 poles identified by ATSS that the Faster R-CNN and RetinaNet methods were unable to detect. In AP75, this number rises to 7811 utility poles located by ATSS that the other two methods were unable to detect.
Round 2
Reviewer 2 Report
The authors have provided a thorougly revised manuscript; all issues raised previously were addressed to my satisfaction, I have no further comments